# A Study of Some Mechanical Properties of a Category of Composites with a Hybrid Matrix and Natural Reinforcements

**DOI:** 10.3390/polym11030478

**Published:** 2019-03-12

**Authors:** Marius Marinel Stănescu, Dumitru Bolcu

**Affiliations:** 1Department of Applied Mathematics, University of Craiova, 13 A.I. Cuza, 200396 Craiova, Romania; mamas1967@gmail.com; 2Department of Mechanics, University of Craiova, 165 Calea Bucureşti, 200620 Craiova, Romania

**Keywords:** composite materials, hybrid resin, natural reinforcement, mechanical properties

## Abstract

The current composite materials must meet a double challenge, one that involves obtaining mechanical properties suitable to the field of activity in which they are used and another one, equally important, that requires that they be renewable. In this paper, we have obtained a category of composite materials that have natural reinforcements (fabrics of flax, cotton, hemp, cattail leaves, and wheat straw). As a matrix, we have used three types of hybrid resin, in the composition of which we used the natural resin dammar, in different majority volume proportions. The differences, up to 100%, were represented by epoxy resin and its associated reinforcement, to generate a quick process of polymerization. We have measured certain mechanical properties and the damping properties of the three types of hybrid resin and of the composite materials under study. Based on these properties, we point out a few fields of activity where these composite materials can be used.

## 1. Introduction

A hybrid resin involves the combination of two constituents: an organic and an inorganic constituent. We have to mention that most attempts at obtaining such resins pertain to the lacquer industry (see [1,2,3]).

Starting from the fact that natural resins cannot make thick resins (see, for instance, [1] and [2]), we may draw the conclusion that the bio-resins studied until now have actually been hybrid resins. Therefore, we will refer below to a selective bibliography concerning this type of research.

Bio-resins are resins derived from a biological source, and consequently, they can be biodegradable and compostable; thus, hypothetically, after they have been utilized, they can be decomposed. Sandarac, copal, and dammar are the most often employed vegetable resins. Natural resins are insoluble in water; however, they are slightly soluble in oil, alcohol, and, partly, in petrol. They form solutions with certain organic solvents, solutions that can be used as covering lacquers. Turpentine, colophony, and mastic are products resulting from the distillation of conifer resins. A study concerning the chemical composition of the properties of these resins was carried out in [4], and the applications were shown in [5].

In [6], mechanical characteristics, characteristics of water vapor transmission, and characteristics of moisture absorption of dammar films were studied. These had been produced by the method of solvent casting/evaporation from solutions, which did not contain a softening agent and which contained a softening agent.

The work [7] shows a new binder, modified from silicon and dammar, which can decrease the use of synthetic binders and features improved and more ecological properties. This binder was put to use to cover some aluminum panels that had been submitted to impact, hardness, tensile, and adherence stresses. The optimal composition ensuring the best properties was determined.

In [8], the author focused on the creep behavior, the rigidity, the modulus of elasticity, and the hardness of the modified dammar-silicon, using nanoindentation testing, and we studied the way in which the dammar addition had contributed to improving the elastic behavior.

The work [9] concentrated on extending the knowledge about the microbiological bio-deterioration of dammar-based lacquers in works of art submitted to unsuitable protection conditions, particularly when exposed for a long time to high levels of relative humidity, suggesting that some microorganisms foster natural deterioration phenomena in the case of dammar lacquer films.

The paper [10] focused on composites that make use of starch-based green resins. Different types of fibers were used, and the stratification structure was modified in order to obtain the desired mechanical properties of the composite.

The reaction to the compression stress of palm tree oil treated with different amounts of dammar resin was analyzed in [11].

The paper [12] studied the effects of the waste, residues, or by products on natural fiber-polymer composites and evaluated the potential of these constituents.

The effect of polymethyl methacrylate (PMMA) on the physical properties of dammar for the application of covering lacquers was investigated in [13]. We found out that dammar, when mixed with PMMA, could be used as a cold covering of laminated steel.

The work [14] looked into the possibility of using tallow and dammar for beeswax as PCM (phase change materials) on concrete buildings. The research was conducted in several steps, beginning with testing the thermal properties of beeswax, tallow, and dammar, continuing with the preparation of the concrete cement containing these materials, and, then, the final testing of the concrete having PCM.

The work [15] presented a new type of green composite, in which preforms of short, rigid, and robust natural fibers are used.

In [16], dammar gum was analyzed as a supplementary material meant to improve thermal conductivity and thermal performance in preparing the material for changing the composite phase.

The work [17] approached the manufacture of bio-composite sandwich structures, when both the resin and the reinforcements were natural. In the same line, the work [18] described a new manufacture process for sandwich-type composites, and the test results were statistically analyzed in order to evaluate the effects and responsiveness of the processing parameters, along with the effects of the environment conditions.

Considerable interest has developed recently in using natural fibers instead of synthetic fibers to make composite materials (see [19,20]). These natural fibers include flax, hemp, jute, sisal, knead, coconut, kapok, bananas, henequen, and many others. Natural fibers proved to be suitable reinforcing materials for composites due to a combination of good mechanical properties and advantages as far as the environment protection was concerned, such as renewability and biodegradability (see [21,22]). The advantages of the natural fibers compared to the traditional fibers include: relatively lower cost, light weight, less damage to the operating equipment, improved surface finishing of the cast pieces (in comparison with steel fiber composites), and good relative mechanical properties.

The main disadvantages of natural fibers are: their processing is limited to low-temperature processing (limited thermal stability); their tendency to form piles; and their hydrophilic nature (see [23,24,25]). The experimental data on their mechanical properties, particularly when they are tested under various processing conditions, have shown inconsistent values in many cases (see [20,21,22]). The irregular characteristics of natural fibers are one of the reasons explaining this. In addition, in many cases, there are problems linked to the interfacial properties of this type of composite (see [20,21,22]). Good results were also obtained through the development of the biopolymer nanocomposites (see, for example, [26,27]), as a successful strategy to obtain green composites with excellent performances.

In this paper, we present some mechanical properties of three sets of samples made of a hybrid resin, in the composition of which we used a dammar volume proportion of 55%, 65%, and 75%, respectively. The difference, up to 100%, consisted of epoxy resin and its associated reinforcement. The properties were determined based on tensile tests. We picked a hybrid resin sample of each set of samples, and we determined its chemical composition by using EDS (energy-dispersive X-ray spectroscopy). We studied the influence of the epoxy resin volume proportion on the mechanical behavior of the hybrid resin. We also determined, based on tensile testing, the characteristic curves, tensile strength, elongation at break, modulus of elasticity, and the images of the break area for the samples made of composite materials with the hybrid resin matrix that we had obtained previously. As for reinforcement, we used fabrics of hemp, cotton, and flax, wheat straws and cattail leaves. In order to expand the applicability of these composite materials, we also looked into the vibration damping properties.

## 2. Materials and Methods

### 2.1. Making the Samples

The natural resin dammar was diluted by turpentine, and if it was kept in closed containers, it stayed in a liquid state. This composition was used for painting (varnish) protection. The disadvantage lies in that the process of resin hardening is very long, even if it is applied in thin layers. We tried to eliminate this shortcoming by adding a reduced proportion of synthetic resin, together with its associated hardener, to generate points of quick activation of the polymerization process.

In the first stage, we cast three hybrid-resin plates, where we used a dammar volume proportion of 55%, 65%, and 75%, respectively. The difference up to 100% consisted of epoxy resin of the Resoltech 1050 type, together with the associated hardener Resoltech 1055. The casting temperature was 21–23 ${}^{\circ }$C. To realize the Resoltech 1050/Resoltech 1055 combination, we respected the manufacturer’s instructions. We used a mixture ratio of 7/3 after the given volume. We mixed the epoxy resin obtained with dammar resin. All samples based on hybrid resin were cut after 10 days.

We cut out three sets of ten samples each, labeled Da 1.1-10, Da 2.1-10, and Da 3.1-10. The sample sizes were: 250 mm long, 25 mm wide, and 6.2 mm thick based on the ASTM D3039 standard [28]. The sample density ranged between 1040 and 1060 kg/m${}^{3}$.

Figure 1 shows hybrid resin samples of each set.

In the second stage, we used hybrid resin with a dammar volume proportion of 65%, and we cast five plates of composite materials, having reinforcement as the following:-14 layers of fabric made of a mixture of 60% flax and 40% cotton (with a specific mass of 240 g/m${}^{2}$); we obtained a composite material with a 60% mass proportion of resin and 1200 kg/m${}^{3}$ density;-28 layers of cotton fabric (with a 126 g/m${}^{2}$ specific mass); we obtained a composite material with a 60% mass proportion of resin and 1180 kg/m${}^{3}$ density;-7 layers of hemp fabric (with a specific mass of 352 g/m${}^{2}$); we obtained a composite material with a 63% mass proportion of resin and 1110 kg/m${}^{3}$ density;-8 almost compact layers of wheat straw; we obtained a composite material with a 59% mass proportion of resin and 1040 kg/m${}^{3}$ density;-8 almost compact layers of cattail leaves; we obtained a composite material with a 58% mass proportion of resin and 1050 kg/m${}^{3}$ density.

We cut out 10 samples of each plate, with the sizes: 250 mm long and 25 mm wide (according to the ASTM D3039 standard [28]). The thickness was: 6.2 mm for the samples reinforced by flax; 6.3 mm for the samples reinforced by cotton; 6.3 mm for the samples reinforced by hemp; 5.0 mm for the samples reinforced by wheat straw; 5.0 mm for the samples reinforced by cattail leaves; DI(hybrid resin-flax fabric); DB(hybrid resin-cotton fabric); DC(hybrid resin-hemp fabric); DPa(hybrid resin-wheat straw); DP(hybrid resin-cattail leaves). Figure 2 shows samples of each set.

The thermo-mechanical properties of Resoltech 1050 epoxy resin, together with its associated hardener Resoltech 1055, were given by the producer (see [29]).

### 2.2. Analysis Methods and Equipment Used

The samples made in Section 2.1 were subjected to the following experimental determinations: tensile test, SEM and EDS analysis, and vibration analysis.

#### 2.2.1. Tensile Test

Figure 3 shows the assemblage of a hybrid resin sample tensile test.

All three types of hybrid resin and the types of composite materials underwent a tensile test, which was carried out according to the ASTM D3039 standard (see [28]). We used the LRX Plus testing machine from LLOYD Instruments.

The elements obtained from this trial were: the characteristic curve, tensile strength $R_{m}$ (MPa), percentage elongation after fracture *A* (%), and elasticity modulus *E* (MPa).

#### 2.2.2. EDS and SEM Analysis

The chemical composition of a hybrid resin sample (the one with a dammar volume proportion of 65%) was measured by EDS analysis.

This analysis was performed with the help of a PHENOM PURE PRO X scanning electron microscope (with integrated EDS), with a conventional cathode geared for the microscopic study of the structure and surface of different materials, with the possibility to determine chemical composition and structure phases. The magnifying power was between 80 and 130,000 times.

The SEM (Scanning Electron Microscope) analysis was performed by an electron microscope, Hitachi model S3400N/type II (see [30] with the technical characteristics given by the manufacturer).

We used this microscope because of its greater magnification range than the microscope of the EDS analysis.

#### 2.2.3. Vibration Analysis

We experimentally determined the damping coefficient and the characteristic frequency both for the samples of the three types of hybrid resin and for the composite materials samples made of hybrid resin, in which the dammar volume proportion was 65% and the reinforcement consisted of flax, cotton, and hemp fabric.

The studied samples were restrained at one end with a jaw vise on a big table, each sample having free lengths of 120 mm, 140 mm, 160 mm, and 180 mm sequentially. At the free end, on the longitudinal axis, the Brüel & Kjaer 8309 accelerometer was glued by wax. The accelerometer is specifically designed for vibration measurements. For each assemblage condition of the sample type and free length, an initial deformation was transmitted to the sample, and by release, the sample performed free damped vibrations. The free damped vibrations were recorded for 5 s approximately. The sample frequency was 2400 Hz/channel.

To study the vibrations, the following measuring equipment was used:-a SPIDER 8 data acquisition system, connected via USB to a notebook;-the data acquisition set was processed by the CATMAN EASY software, which linked the two entities;-a NEXUS 2692-A-0I4 conditioning amplifier connected to a SPIDER 8 system;-an accelerometer with a 0.04 pC/ms${}^{-2}$ sensitivity, connected to the conditioning amplifier, where C means Coulomb, pC represents picocoulomb ($10^{-12}$ C ), and ms${}^{-2}$ represents the acceleration measurement unit.

The frequency measurement range was set from 0–2400 Hz in SPIDER 8. To eliminate the errors introduced by the experimental system, we performed a Butterworth “high pass” filtration at a 3-Hz frequency for each measurement.

In order to determine the system damping, we used the logarithmic decrement method.

The experimental recording of the free vibrations in a certain point enabled the calculation of the damping factor μ with the help of the formula (see [31]):$$\mu =\frac{1}{t_{2}-t_{1}}\ln\frac{w_{1}}{w_{2}}$$
where:-$t_{1}$ and $t_{2}$ are the time values for two successive peaks of the amplitude diagram;-$w_{1}$ is the peak of amplitude at $t_{1}$, and $w_{2}$ is the peak of amplitude at $t_{2}$.

After determining the damping factor μ, the damping curve was drawn $f\left(t\right)=w_{0}\exp\left(-\mu t\right)$, for which the program overlayed the recorded characteristic, where $w_{0}$ represents the initial amplitude of the free damped vibrations. Since the damping of the plates in free damped vibrations is a combination of structural damping and damping due to air friction, in all recordings, in order to determine the damping factor μ, we selected areas where the vibration damping ranged between the amplitudes of $w_{1}=2$ mm and $w_{2}=0.05$ mm.

A spectral analysis was carried out by using FFT (fast Fourier transform) techniques, and this determined the free damped oscillation frequency.

## 3. Results

By a representative sample of a set, we will understand the sample for which the experimental results are close to the mean value (arithmetic mean) of the studied mechanical properties for the whole set of samples of that type.

### 3.1. The Experimental Results for the Hybrid Resins Studied

In this subsection, we present the results we obtained for the three types of hybrid-resin samples (whose method of fabrication was described in Section 2.1).

In Figure 4, Figure 5 and Figure 6, we show the characteristic curves of several representative hybrid resin samples where we used a dammar volume percentage of 55% (Da 1.x), 65% (Da 2.x), and 75% (Da 3.x), respectively.

Tensile strength $R_{m}=21.4$ MPa; elongation at break A=2.06%; modulus of elasticity E=1332 MPa.

Tensile strength $R_{m}=16.3$ MPa; elongation at break A=2.98%; modulus of elasticity E=799 MPa.

Tensile strength $R_{m}=8$ MPa; elongation at break A=4.16%; modulus of elasticity E=589 MPa.

In Table 1, we show the value limits recorded for the modulus of elasticity *E* (MPa), the tensile test $R_{m}$ (MPa), and the elongation at break *A* (%), for the sample set of the Da 1.x, Da 2.x, and Da 3.x type.

In the following, we make the EDS analysis for the obtained resin samples to evidence the chemical composition and the SEM analysis to observe the structure changes produced by the variation of the mass proportion by dammar.

Figure 7 shows the images resulting from SEM analysis of the three hybrid resin types and epoxy resin.

From the SEM analysis of the three types of hybrid resins, we noticed that a higher dammar volume proportion generated a higher number of air bubbles. A possible explanation for this may be that the hardening time and, implicitly, the duration of the polymerization process increased with the dammar volume proportion in the hybrid resin. The hardening reaction took place with the release of bubbles of air that were eliminated through the upper surface of the plates. During the polymerization process, the viscosity of the resin increased, and the air bubbles generated in the last part of the reaction remained captive inside the resin.

Based on the EDS analysis of a hybrid resin specimen, taken from the Da 2.x samples, we show in Figure 8 the diagram of the chemical composition obtained at a 15-keV intensity.

In Table 2, we show the chemical composition of a hybrid-resin specimen taken from the Da 2.x sample, obtained from the EDS analysis.

The presence of chemical elements in the structure of this hybrid resin is expressed by atomic concentration (for an element X) and by weight concentration. These are given by the formulas:$$\mathrm{Atomic}\;\mathrm{Concentration}\;=\frac{\mathrm{number}\;\mathrm{of}\;\mathrm{atoms}\;\mathrm{of}\;\mathrm{element}\;X}{\mathrm{total}\;\mathrm{number}\;\mathrm{of}\;\mathrm{atoms}}\cdot 100\%$$
and
$$\mathrm{Weight}\;\mathrm{Concentration}\;=\frac{\mathrm{weight}\;\mathrm{of}\;\mathrm{element}}{\mathrm{total}\;\mathrm{weight}}\cdot 100\%.$$

From Table 2, we see that the main elements in the chemical structure of the hybrid resin Da 2.x type are carbon and oxygen. Therefore, in Table 3, we present the evolution of the atomic concentration (Atomic Conc.) and the weight concentration (Weight Conc.) for the main elements, carbon and oxygen, which appear in the composition of the three types of resin.

We may notice from the previous table that:-in the carbon case, there is a decrease in the atomic concentration and in the weight concentration as the dammar volume proportion is increased;-in the case of oxygen, there is an increase in the atomic concentration and in the weight concentration as the dammar volume proportion is increased.

In Figure 9, we show the vibration recording (characteristic frequency and damping factor) in a sample of the Da 2.x set, for the free length of 120 mm.

Since in Section 3.2, the composite materials to be studied have the type of hybrid resin with a dammar volume proportion of 65% as a matrix, the recording in Figure 9 was made for a sample from the Da 2.x set.

In Table 4, we show the vibration behavior of the epoxy resin samples and of the three types of hybrid resin samples having a width of 25 mm. The values shown represent the arithmetic mean for three measurements.

### 3.2. The Experimental Results for the Composite Materials Studied

In this subsection, we show the results obtained for the composite material samples (whose fabrication method was described in Section 2.1), which were subject to the following experimental determinations:-tensile test;-vibration analysis.

The mechanical properties of the natural fibers used for the fabrics, utilized as reinforcement, are shown in Table 5. We should mention that, in the relevant literature, these properties have variations, depending on their origin, the type of cultivated plants, and the weather conditions in the harvest year, among others. The authors were unable to find mechanical properties of wheat straw and cattail leaves.

We continue by showing, in Figure 10, Figure 11, Figure 12, Figure 13 and Figure 14, the characteristic curves of several representative samples of composite materials with hybrid resin matrix (65% dammar volume proportion) and reinforcements from fabric of flax or hemp fibers, as well as wheat straw and cattail leaves.

We also give the vibration behavior of the composite material samples with reinforcements of flax, cotton, and hemp fabric.

Figure 10 shows the characteristic curve of a representative sample of composite material with flax fabric reinforcement.

Tensile strength $R_{m}=72.0$ MPa; elongation at break A=3.3%; modulus of elasticity E=5072 MPa.

Figure 11 shows the characteristic curve of a representative specimen of cotton fabric-reinforced composite material.

Tensile strength $R_{m}=63.0$ MPa; elongation at break A=8.4%; modulus of elasticity E=3415 MPa.

Figure 12 presents the characteristic curve for a representative sample of hemp fabric-reinforced composite material.

Tensile strength $R_{m}=73.0$ MPa; elongation at break A=2.4%; modulus of elasticity E=6687 MPa.

Figure 13 shows the characteristic curve of a representative sample of wheat straw-reinforced composite material.

Tensile strength $R_{m}=38.0$ MPa; elongation at break A=1.2%; modulus of elasticity E=8370 MPa.

Figure 14 presents the characteristic curve of a representative sample of cattail leaves-reinforced composite material.

Tensile strength $R_{m}=25.0$ MPa; elongation at break A=0.88%; modulus of elasticity E=2911 MPa.

In Table 6, we present the limits of the recorded values for the modulus of elasticity *E* (MPa), the tensile strength $R_{m}$ (MPa), and the elongation at break *A* (%) for the sample sets of these composite materials.

Figure 15 shows the image of the breaking section for a representative specimen (sample) of composite material with the hybrid resin matrix (dammar 65%) and reinforced with wheat straw. It is possible to observe the layered distribution of the composite and highlight the parenchymal tissue provided with large spaces filled with air from wheat straw.

Figure 16 shows the vibration recording (the characteristic frequency and damping factor) for a sample of the composite material of the hemp fabric set, for the free length of 120 mm.

Table 7 shows the vibration behavior of the composite material sample with flax, cotton, and hemp fabric reinforcements, having a width of 25 mm. The values shown represent the arithmetic mean of three measurements.

## 4. Discussion and Conclusions

Comparing the experimental results shows an important change in the mechanical properties when the proportion between dammar and epoxy resin was changed. We noticed a decrease in the values of the tensile strength and the modulus of elasticity as the dammar volume proportion was increased in the mixture. Although the mixtures with a higher epoxy resin quantity had superior mechanical properties, we cannot state that there was a proportionality between the tensile strength or the modulus of elasticity and the epoxy resin volume proportion. Thus, for the 55% dammar mixture, we had the highest values of the mechanical properties. For the 65% dammar mixture, where the epoxy resin volume proportion dropped by 22%, the tensile strength decreased by 24% and the modulus of elasticity decreased by 40%. For the 75% dammar mixture, the epoxy resin volume proportion dropped by 44%, the tensile strength decreased by 63% and the modulus of elasticity by 56%.

The increase of the air volume in the hybrid resin, together with an increase of the dammar volume proportion may explain this decrease. Another explanation might be that the higher proportion of dammar simply weakened the hardened matrix.

Changes also appeared in the forms of the characteristic curves. If in the 55% dammar mixture, the characteristic curve was almost linear, in the 75% dammar mixture, there was an obvious nonlinearity of the characteristic curve.

Changes appeared in the elongation at break as well, which was approximately 2% in the mixture with 55% dammar, 3% in the mixture with 65% dammar, and 4% in the mixture with 75% dammar. This also points to a weaker, more elastic system.

The analysis of the characteristic curves shows a difference in behavior depending on the reinforcing materials. The elongation at break of the composites was comparable with the elongation at break of the reinforcing materials.

The composites reinforced with fabric of flax or hemp fibers (which had elongations at break below 4%) had a low elongation at break, of almost 8.5%.

In the case of the composites with wheat straw or cattail, the elongation at break was very low, around 1%, and at their break, the breaking stress for the resin was reached as well, which yielded in turn. Actually, the characteristic curves of these materials were almost linear as well, linearity being obvious in the cattail leaves-reinforced materials.

If in the case of cattail leaves-reinforced composite materials, we obtained properties comparable to those of the matrix, and in the case of the wheat straw-reinforced composites, we noticed a significant increase in the modulus of elasticity, which was higher than in all the studied materials.

In the case of the composite materials reinforced with cotton fabric, three zones of the characteristic curve can be highlighted. In the first zone, there was a linear dependence between stress and strain, as validated by Hooke’s law. In this zone, the composite behaved as a unit, the fibers and the matrix taking over the stresses together. The second zone was characterized by a significant nonlinearity that was due to important deformations appearing in the matrix; the matrix began to yield, and the stress was transferred to the fibers. In the third zone, linearity reappeared, explained by the fact that the stresses were mostly taken over by the longitudinal fibers, the stress going up to the break of the fibers.

In the samples reinforced with hemp fabric, where the elongation at break of the reinforcing material was small, the nonlinearity of the characteristic curve was smaller, although it had been manifest since the beginning of the stress. This can be explained by the fact that the matrix and the reinforcing material, together, took over the external load during the whole stress time.

In the case of the samples of flax fabric-reinforced composite materials, the behavior of the composite material had an intermediary character; more precisely, the characteristic curve displayed three zones, like in the case of the cotton-reinforced samples; however, the nonlinearities were present in all the zones, as in the hemp-reinforced samples.

The damping capacity is given by the η loss factor. In the study of damped vibrations, the complex modulus of elasticity was used:$$E^{*}=E\left(1+i\eta \right).$$

Under these conditions, the loss factor η is a property of the material, in the same way as the modulus of elasticity, tensile strength, and others.

Since the damping factor depends on the sample length, we may calculate the loss factor (calculated with the ratio $\eta =\frac{\mu }{\pi v}$ given in [35], where μ and *v* are the damping factor and the frequency, according to Table 4 and Table 7) for each of the materials. The following average results were obtained:-for epoxy resin η=0.0204;-for 55% dammar hybrid resin η=0.0642;-for 65% dammar hybrid resin η=0.1052;-for 75% dammar hybrid resin η=0.1355;-for flax fabric-reinforced composites η=0.0743;-for cotton fabric-reinforced composites η=0.0846;-for hemp fabric-reinforced composites η=0.0543.

We may notice that the loss factor decreased as the bar rigidity increased, which is directly proportional to the modulus of elasticity of the material.

Concerning the behavior of sample on vibration, we noticed an increase in the damping capacity (given by η), when the dammar proportion increased in the composition.

If we compare the damping factor value for hybrid resin with 65% dammar resin and the values of the composite materials reinforced with fabric of flax, cotton, or hemp, we realize that there were no significant differences.

In the case of the loss factor, we saw a decrease of the obtained values for the composite materials, compared to hybrid resin with 65% dammar resin. More precisely, the lowest decrease was in the case of cotton fabric-reinforced composites, and the highest was in the case of hemp fabric-reinforced composites.

The mechanical properties, together with the vibration damping properties obtained for the studied composites, recommend them to be used for:-making reusable devices to immobilize fractures, more precisely for making one-size-fits-all pieces that can be fixed afterwards by a self-adhesive bandage system;-making wainscoting or “almost environment-friendly” parquet blocks (as an alternative to wood and PVC);-making reusable formworks for some construction elements.

## Figures and Tables

**Figure 1 polymers-11-00478-f001:**
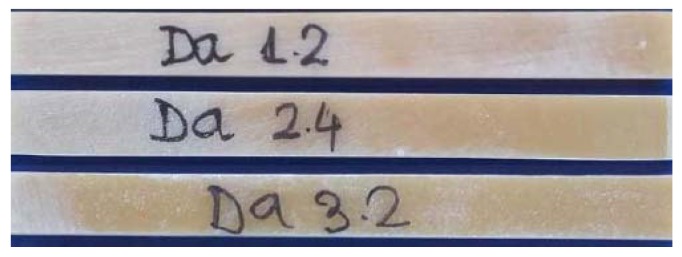
Samples of each set.

**Figure 2 polymers-11-00478-f002:**
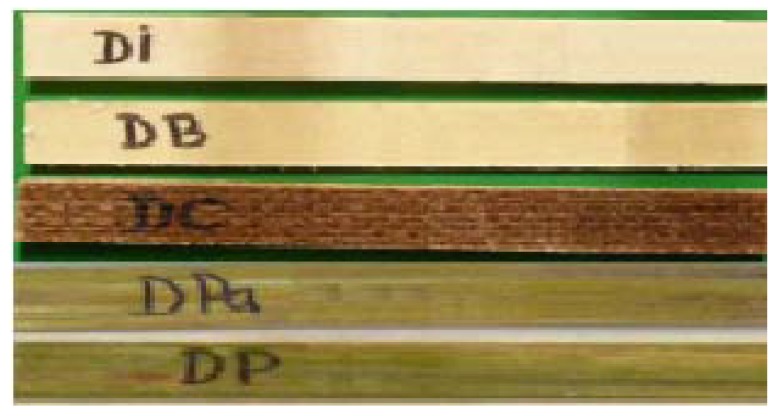
Samples obtained of hybrid resin with 65% dammar proportion, reinforced with fabrics of flax (marked with DI), cotton (marked with DB), hemp (marked with DC), wheat straw (marked with DPa), and cattail leaves (marked with DP).

**Figure 3 polymers-11-00478-f003:**
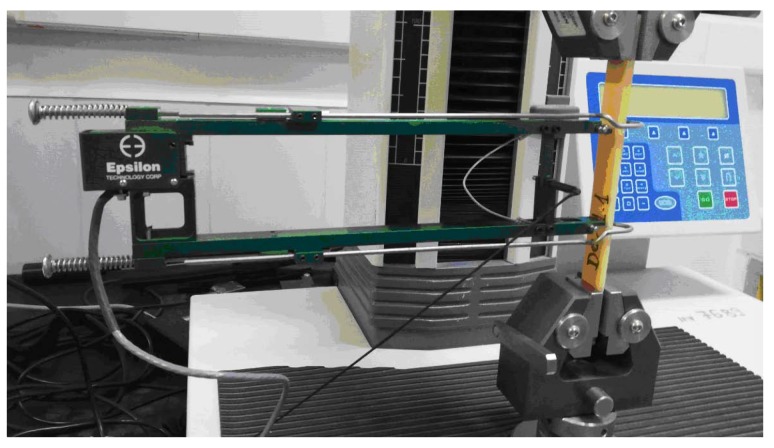
The tensile test assemblage of a sample from the Da 1.x set.

**Figure 4 polymers-11-00478-f004:**
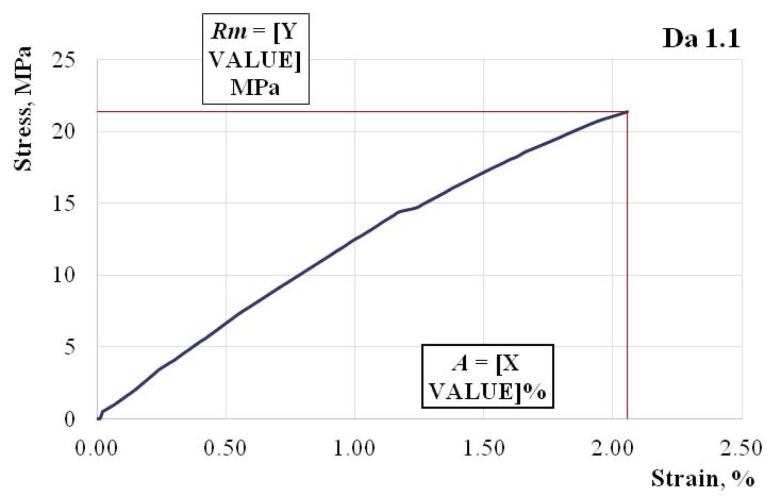
The characteristic curve of a Da 1.x representative sample.

**Figure 5 polymers-11-00478-f005:**
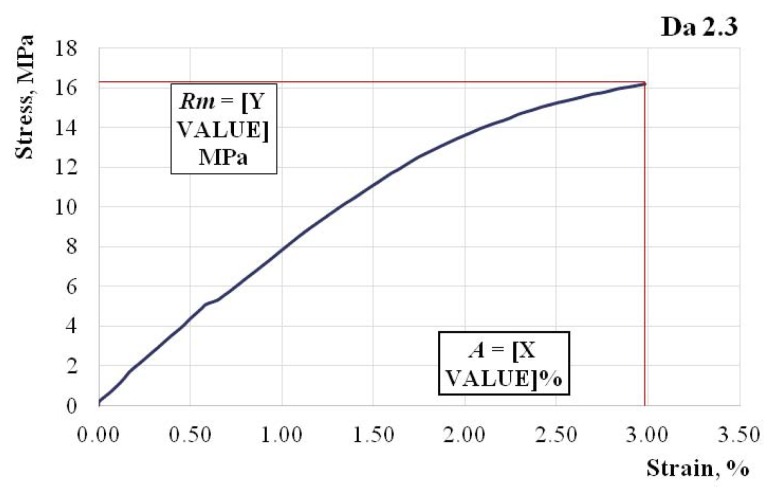
The characteristic curve of a Da 2.x representative sample.

**Figure 6 polymers-11-00478-f006:**
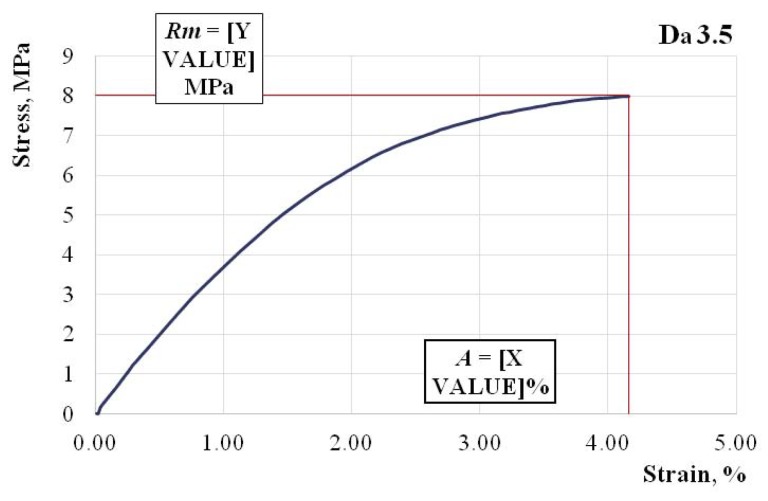
The characteristic curve of a Da 3.x representative sample.

**Figure 7 polymers-11-00478-f007:**
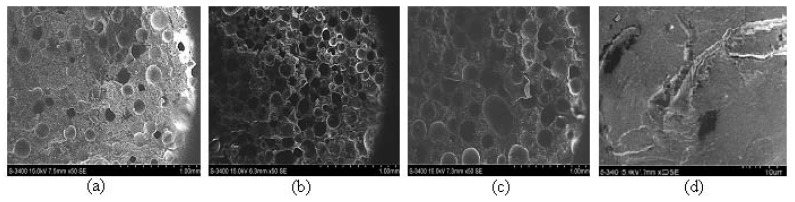
SEM analysis for a sample of the type: (**a**) Da 1.x. (**b**) Da 2.x. (**c**) Da 3.x. (**d**) epoxy resin.

**Figure 8 polymers-11-00478-f008:**
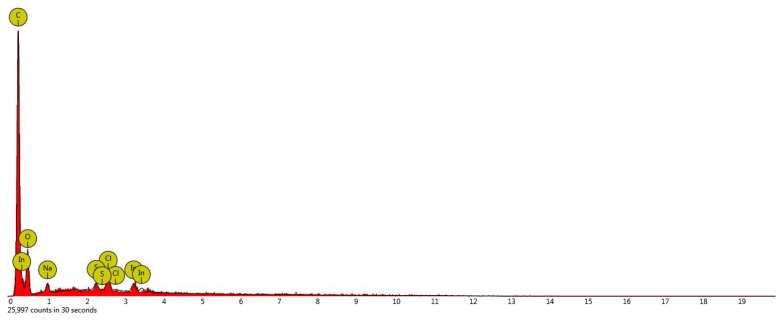
The EDS analysis diagram of the chemical composition of a hybrid-resin specimen taken from the Da 2.x sample, obtained at a 15-keV intensity.

**Figure 9 polymers-11-00478-f009:**
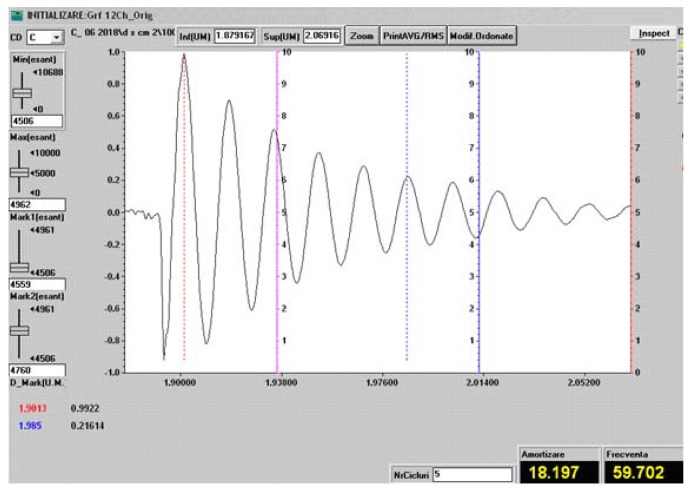
Vibration recording (the characteristic frequency and damping factor) in a sample of the Da 2.x set, for the free length of 120 mm.

**Figure 10 polymers-11-00478-f010:**
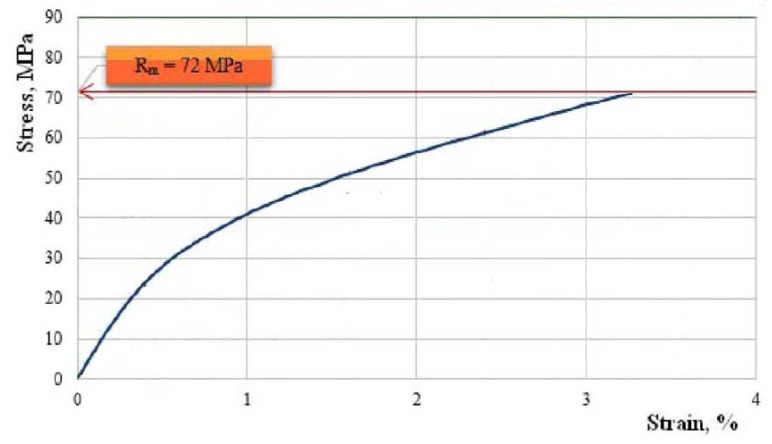
The characteristic curve of a representative sample of composite material with flax fabric reinforcement.

**Figure 11 polymers-11-00478-f011:**
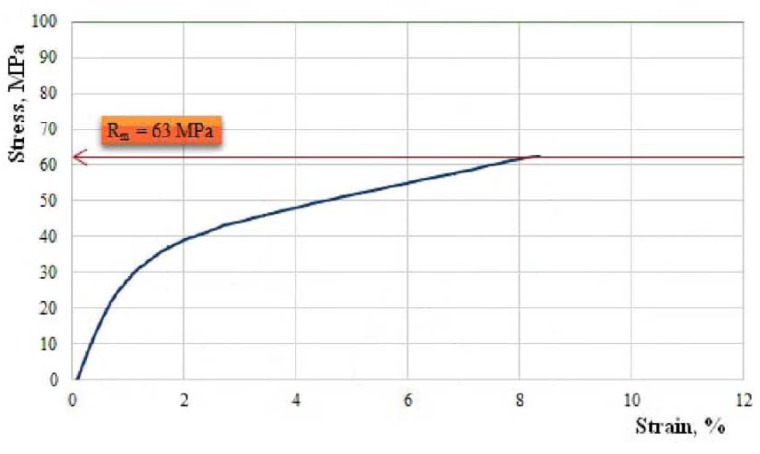
The characteristic curve of a representative sample of cotton fabric-reinforced composite material.

**Figure 12 polymers-11-00478-f012:**
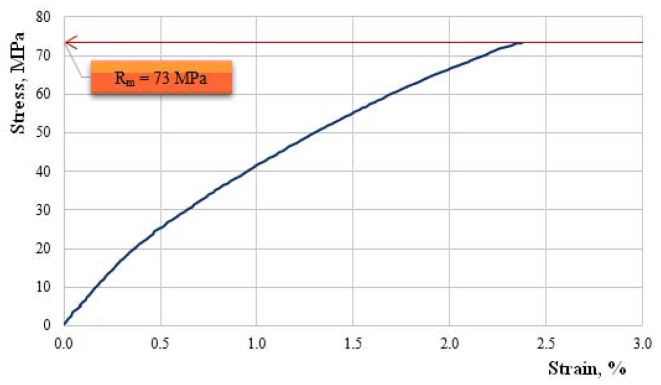
The characteristic curve for a representative sample of hemp fabric-reinforced composite material.

**Figure 13 polymers-11-00478-f013:**
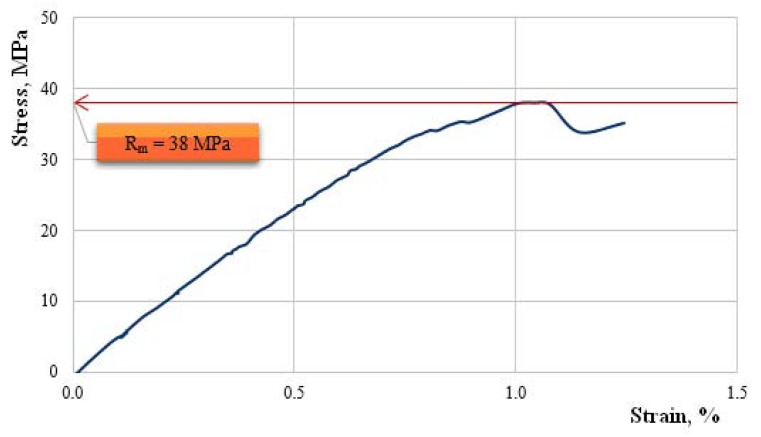
The characteristic curve of a representative sample of wheat straw-reinforced composite material.

**Figure 14 polymers-11-00478-f014:**
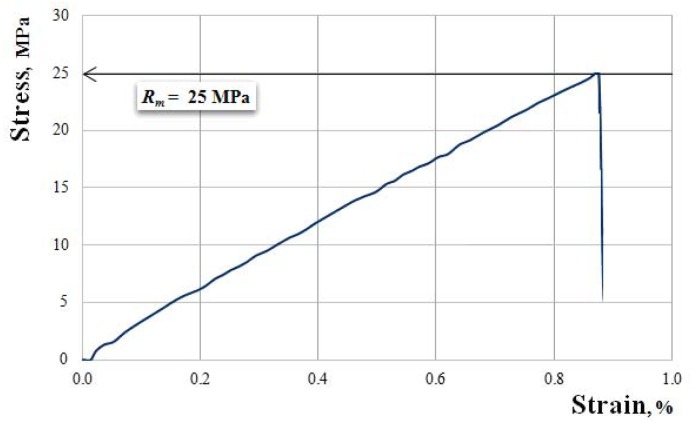
The characteristic curve of a representative sample of cattail leaves-reinforced composite material.

**Figure 15 polymers-11-00478-f015:**
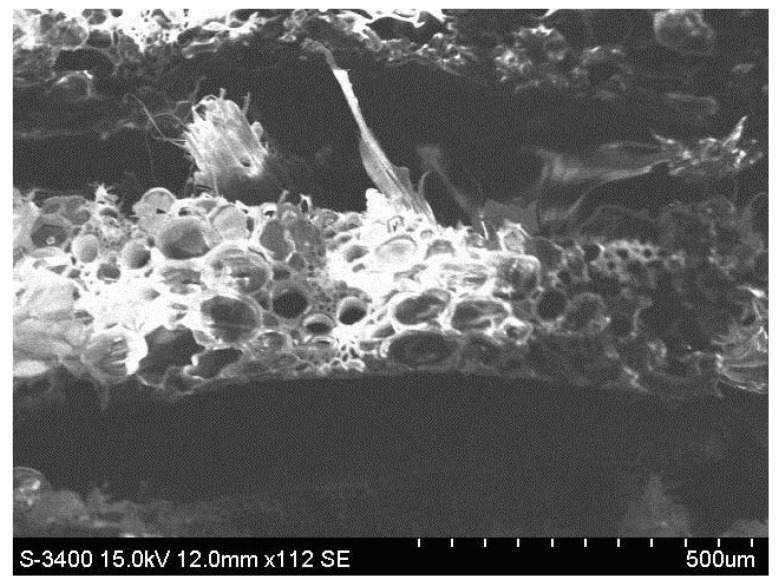
The image of the breaking section of a representative sample of wheat straw-reinforced composite material.

**Figure 16 polymers-11-00478-f016:**
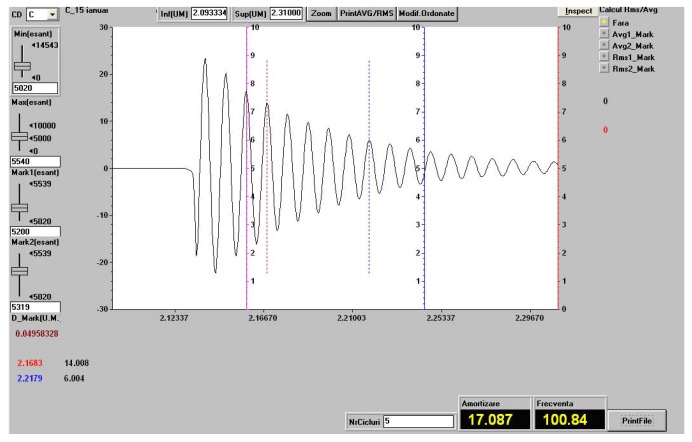
The vibration recording (the characteristic frequency and the damping factor) for a composite sample of the hemp fabric set, for the free length of 120 mm.

**Table 1 polymers-11-00478-t001:** The value limits of the modulus of elasticity, tensile strength, and elongation at break for the hybrid resin samples.

Sample Type	Modulus of Elasticity	Tensile Strength	Elongation at Break
	*E* (MPa)	$R_{m}$ (MPa)	*A* (%)
Da 1.x	1260–1410	20.4–21.8	1.84–2.06
Da 2.x	750–860	15.9–16.7	2.85–3.17
Da 3.x	560–630	7.7–8.2	3.76–4.56
Resoltech 1050	3550–3600	53.9–54.3	1.8–1.9
(hardener Resoltech 1055)			

**Table 2 polymers-11-00478-t002:** The chemical composition of a hybrid resin sample, taken from the Da 2.x sample. Conc., concentration.

Element	Element	Element	Atomic	Weight
Number	Symbol	Name	Conc.	Conc.
6	C	Carbon	77.74	69.26
8	O	Oxygen	20.47	24.30
49	In	Indium	0.40	3.37
17	Cl	Chlorine	0.59	1.56
11	Na	Sodium	0.59	1.01
16	S	Sulfur	0.21	0.50

**Table 3 polymers-11-00478-t003:** The evolution of the carbon and oxygen in the composition of the three types of resin.

Resin Type	Carbon (C)		Oxygen (O)	
	Atomic	Weight	Atomic	Weight
	Conc.	Conc.	Conc.	Conc.
Da 1.x	87.88	84.48	12.12	15.52
Da 2.x	77.74	69.26	20.47	24.30
Da 3.x	66.89	63.23	25.81	28.60

**Table 4 polymers-11-00478-t004:** The vibration behavior of (epoxy and dammar) resin samples.

Free Length	Epoxy Resin		Da 1.x		Da 2.x		Da 3.x	
	Frequency	Damping	Frequency	Damping	Frequency	Damping	Frequency	Damping
	ν (Hz)	μ (s${}^{-1}$)	ν (Hz)	μ (s${}^{-1}$)	ν (Hz)	μ (s${}^{-1}$)	ν (Hz)	μ (s${}^{-1}$)
120	120.05	8.22	76.85	13.02	59.70	18.04	52.48	21.04
140	88.52	5.71	56.02	11.26	44.28	14.60	38.22	15.63
160	68.11	4.08	43.25	9.14	34.68	11.38	28.31	12.02
180	52.98	3.22	33.86	7.62	26.91	9.72	21.85	10.22

**Table 5 polymers-11-00478-t005:** The main mechanical properties of the utilized fibers (see for example [23,32,33,34]).

Fibers	Density	Elongation at Break	Tensile strength	Modulus of Elasticity
	ρ (kg/m${}^{3}$)	*A* (%)	$R_{m}$ (MPa)	*E* (MPa)
Cotton	1500–1600	7.0–8.0	5.5–12.6	287–800
Flax	1500	2.7–3.2	27–39	345–1100
Hemp	1400–1500	2–4	30–60	310–750

**Table 6 polymers-11-00478-t006:** The value limits of the modulus of elasticity, tensile strength, and elongation at break for the composite material samples studied.

Sample Type	Modulus of Elasticity	Tensile Strength	Elongation at Break
	*E* (MPa)	$R_{m}$ (MPa)	*A* (%)
Dammar Flax	5000–5220	71–74	3.2–3.5
Dammar Cotton	3050–3420	63–67	8.4–9.5
Dammar Hemp	6350–6780	72–75	2.2–2.4
Dammar Wheat Straw	7450–8480	33–39	1.0–1.3
Dammar Cattail Leaves	2610–2950	22–25	0.85–0.95

**Table 7 polymers-11-00478-t007:** The vibration behavior of the composite material specimens.

Free Length	Flax Fabric		Cotton Fabric		Hemp Fabric	
	Reinforced Composite		Reinforced Composite		Reinforced Composite	
	Frequency	Damping	Frequency	Damping	Frequency	Damping
	ν (Hz)	μ (s${}^{-1}$)	ν (Hz)	μ (s${}^{-1}$)	ν (Hz)	μ (s${}^{-1}$)
120	87.59	22.42	64.86	17.00	100.21	17.11
140	67.89	17.37	51.28	13.83	75.47	12.72
160	54.29	11.09	40.61	10.05	57.74	9.40
180	42.47	9.23	31.29	8.88	45.37	8.18

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
