# Peer review of "A Study of Some Mechanical Properties of a Category of Composites with a Hybrid Matrix and Natural Reinforcements"

_polymers, 2019, doi:10.3390/polym11030478_

Round 1

Reviewer 1 Report

The revised version is much better than the original one. However, there are still some suggestions for the authors:

1. Please avoid using the first person (e.g., “we”) in the paper.

2. There are so many paragraphs in the paper that contain only one sentence. Please combine them into larger paragraphs.

3. Page 1, Lines 26-30: The whole paragraph is a grammatically wrong sentence, and it was still not revised from the first version.

4. Page 4, Line 119: Please mention the ASTM standard in your text and it does not need to be one of the references.

5. Page 5, Lines 179-185, including Fig. 3: These are not results and should not appear in Section 3. Please move this part to Section 2.

6. Page 10, Figs. 10 and 11: Why there are two curves in both figures while Figs. 12-14 only have one curve? What does Rp0,2 mean in these two figures?

Grammar errors:

1. Page 3, Line 90: “As reinforcement, we used fabrics of hemp, …” should be “As for reinforcement, we used fabrics of hemp, …”.

2. Page 13, Line 294: “The increase in the air volume in the hybrid resin, together with an increase in the Dammar 294 volume proportion may explain this decrease.” Should be “The increase of the air volume in the hybrid resin, together with an increase of the Dammar 294 volume proportion may explain this decrease.”

Author Response

Dear Reviewer 1,

In accordance with your last remarks, we have made the following changes in the article structure:

Title: “A STUDY OF SOME MECHANICAL PROPERTIES OF A CATEGORY OF COMPOSITES WITH A HYBRID MATRIX AND NATURAL REINFORCEMENTS

Authors: Marius Marinel Stănescu and Dumitru Bolcu

1. Please avoid using the first person (e.g., “we”) in the paper.

We have corrected this error. We gave our manuscript at the Department of Anglo-American Studies (Faculty of Letters – University of Craiova) for check by the native speakers of English.

2. There are so many paragraphs in the paper that contain only one sentence. Please combine them into larger paragraphs.

We have respected the template/structure of a paper for MDPI / Polymers. We have made the moves required from one paragraph to the next.

3. Page 1, Lines 26-30: The whole paragraph is a grammatically wrong sentence, and it was still not revised from the first version.

We have corrected the entire paper (from grammatically point of view - including the indicated lines).

4. Page 4, Line 119: Please mention the ASTM standard in your text and it does not need to be one of the references.

The other reviewer explicitly requested the introduction of ASTM D3039 in the bibliography (if you ask for this we will remove ASTM from the bibliography).

5. Page 5, Lines 179-185, including Fig. 3: These are not results and should not appear in Section 3. Please move this part to Section 2.

We moved the indicated lines at the beginning of subsection 2.2.

6. Page 10, Figs. 10 and 11: Why there are two curves in both figures while Figs. 12-14 only have one curve? What does Rp0,2 mean in these two figures?

For materials with a high elongation at break, we determined Rp0.2, which is the tension at which a specific remainder deformation of 0.2% is obtained after the removal of the external load. This stress is highlighted in materials having a high elongation at which remainder plastic deformations. For materials with low elongation at break with a breakable behavior (where breakage suddenly appears) Rp0.2 not be highlighted.

Grammar errors:

1. Page 3, Line 90: “As reinforcement, we used fabrics of hemp, …” should be “As for reinforcement, we used fabrics of hemp, …”.

We have fixed the error.

2. Page 13, Line 294: “The increase in the air volume in the hybrid resin, together with an increase in the Dammar 294 volume proportion may explain this decrease.” Should be “The increase of the air volume in the hybrid resin, together with an increase of the Dammar 294 volume proportion may explain this decrease.”

We have fixed the error.

We have marked the changes:

-        with red-colour the answers for reviewer 1;

-        with green-colour the answers for reviewer 2;

-        with blue-colour the answers for both reviewers and grammatically corrections;

in the paper's content - attached to this answer.

We thank you for your viewpoints based on which we have made the necessary changes to increase the scientific level of the article.

                                                                                                               The authors

Reviewer 2 Report

The paper has improved, but many comments and requests of the first Review have not been followed.

Author Response

Dear Reviewer 2,

In accordance with your last remarks, we have made the following changes in the article structure:

Title: “A STUDY OF SOME MECHANICAL PROPERTIES OF A CATEGORY OF COMPOSITES WITH A HYBRID MATRIX AND NATURAL REINFORCEMENTS

Authors: Marius Marinel Stănescu and Dumitru Bolcu

Line 13 Comments: “Hybrid resin” is a general      term and can be also the combination of two organic components, as this is      usually the case in adhesive application.

Answer: We used the term "hybrid resin" because it contains a natural resin (Dammar) and epoxy resin (which is synthetic). We believe that the resin studied by us may fall within the term hybrid resin, which contains a wide range of resins. Sure there may be the combinations of two or more natural resins, and in this case the result will be a hybrid bio-resin. Based on existing studies (see for example [Kanehashi S.] and [Ishimura T.]), do we believe that the bio resin can not be strengthened without a synthetic hardener?

Line 16 Comments: There are many natural resins,      which show quite high viscosity, such as impurified tannins with a certain      portion of high molar mass carbohydrates. The term “natural resins” can be      used for many substances; it should be therefore explained clearly what is      the meaning of this term in this paper.

Answer: Natural resins come from nature without chemical processing. Even if some of them have a high viscosity, for strengthen, is required the combination with a siccative (hardening agent).

Line 23 Comments: “… colophony, and mastic are …”

Answer: We have made the modification "... colophony, and mastic are ...".

Line 104 ff Comments: Use SI units. Means “kg/m³”      instead of “g/cm³”.

Answer: We put “kg/m³” instead of “g/cm³”.

Line 108 ff Comments: Please express the mass      proportion in % (see first review).

Answer: We have replaced the mass ratio in "%".

Line 122-124 Comments: Explanations should be put      to the caption of Figure 2.

Answer: We have put in the legend of Figure 2 the required explanations.

Line 199-202 and 294 Comments: What is the source      of the air? Why is the volume of air higher with higher proportion of      Dammar? What can be the influence of the duration of polymerization? See      also 161 of first review - Bigger air bubbles mean that there is more      volume decrease when curing takes place. This would mean that with higher      percentage of Dammar such a stronger shrinkage is given. However, the      range in variation of the Dammar proportion is quite narrow, only 55 –      75%. What is the picture with pure epoxy? Is it possible to have pure Dammar?      Is it curing at all? If yes, under which conditions?  What is missing, are these two corner      stones of the pure materials.

Answer: The hardening reaction takes place with release of bubbles of air that are eliminated through the upper surface of the plates.  During the polymerization process the viscosity of the resin increases and the air bubbles generated in the last part of reaction remain captive inside the resin. This process is highlighted by SEM analysis. We added a figure with SEM analysis of an epoxy resin specimen. Based on existing studies (see for example [Kanehashi S.] and [Ishimura T.]), do we believe that the bio resin can not be strengthened without a synthetic hardener?

Line 207 Comments: Please define “Atomic Conc.”      and “Weight Conc.”

Answer: We have defined “Atomic Conc.” and “Weight Conc.”.

Line 215 Comments: What is the logical connection      between SEM and the concentration of the two elements C and O? Please      explain better what the sentence in line 215 is telling.

Answer: We do not have a logical explanation between the SEM analysis and the concentration of the two elements C and O. We did the EDS analysis to highlight the chemical composition and SEM analysis to observe the structural changes caused by the variation in the mass ratio of the Dammar.

Line 216-227 Comments: These lines belong to      Chapter 2 Materials and methods.

Answer: We have put these lines at the end of section 2 "Materials and Methods".

Line 245 Comments: “so on and so forth” needs to      be replaced by detailed parameters; add some more parameters and then you      might write “among others”.

Answer: We have made the change.

Figure 10 ff Comments: What is Rp0.2?

Answer: For materials with a high elongation at break, we determined Rp0.2, which is the tension at which a specific remainder deformation of 0.2% is obtained after the removal of the external load. This stress is highlighted in materials having a high elongation at which remainder plastic deformations. For materials with low elongation at break with a breakable behavior (where breakage suddenly appears) Rp0.2 not be highlighted.

Line 276 Comments: What can be seen in Figure 15 (see also first      review).

Answer: Figure 15 shows the image of the breaking section for a representative specimen (sample) of composite material with the hybrid resin matrix (Dammar 65%) and reinforced with wheat straw. It is possible to observe the layered distribution of the composite and highlight the parenchymal tissue provided with large spaces filled with air from wheat straw.

Table 6 Comments: Give some results of other composite      materials for sake of comparison, such as composite materials based on      synthetic matrix and synthetic fibbers.

Answer: A comparison between the composites studied by us and the composites based on synthetic resins and fibers is not conclusive, because synthetic composites have superior elastic and resistance properties, which can be controlled by manufacturing. Composites with hybrids resins and natural reinforcements have the advantage of being "environmentally friendly", and the mechanical properties recommend them for use in areas where there is not much mechanical stress.

Figure 3 Comments: Belongs to chapter 2 (see      first review).

Answer: We have moved the figure in Section 2.2.

Line 100 ff Comments: What had been the      conditions during casting? Temperature, mixing ratio between epoxy resin      and hardener? Time until casting samples were finally cured?

Answer: The casting temperature was 21-230C. To realize the Resoltech 1050/Resoltech 1055 combination we respected the manufacturer’s instruction. We used a mixture ratio of 7/3 after given volume. We mixed the epoxy resin obtained with Dammar resin. All samples based on hybrid resin were cut after 10 days.

Line 295 Comments: Another explanation might be      that the higher proportion of Dammar simply weakens the hardened matrix.

Answer: We have completed with the suggested explanation.

Line 300 Comments: This also points to a weaker,      more elastic system.

Answer: We have completed with the suggested explanation.

Line 348-350 Comments: This is clear and the task      of reinforcement. Please delete this sentence.

Answer: We delete this sentence.

General remarks (see partly first review):

Comparison with pure epoxy is missing.

Answer: We have put in table the properties values of combination Resoltech 1050/Resoltech 1055.

No information is given on how the epoxy resin      and the Dammar will react, if at all. Are they hardening separately only      such as two interpenetrating networks? Is there any chemical reaction      between the two substances? Will it be probable that there might be a      reaction?

Answer: Until this moment, there are not researches strictly about Dammar resin or Dammar in combination with another type of resin. In the future, we will collaborate with „Institut für Anorganische und Analytische Chemie – Technische Universität Braunschweig, Deutschland”, to analyze the chemical structure and not only Dammar natural resin, and eventually to recommend a siccative.

Are there any quality requirements given by      standards for such composites? Will the Dammar composites fulfil these      requirements?

Answer: There are no explicit quality standards for such composites with natural fabric reinforcement and "hybrid" matrix (based on natural resins: Dammar, Copal of Manila, Sandarac and others).

Had various physical properties been tested, such      as behaviour in contact with water? Thickness swelling? Water absorption?

Answer: In the future, we will also study these properties.

Check of language belongs to the Publishers.

Answer: We gave our manuscript at the Department of Anglo-American Studies (Faculty of Letters – University  of Craiova) for check by the native speakers of English.

We have marked the changes:

-        with red-colour the answers for reviewer 1;

-        with green-colour the answers for reviewer 2;

-        with blue-colour the answers for both reviewers and grammatically corrections;

in the paper's content - attached to this answer.

We thank you for your viewpoints based on which we have made the necessary changes to increase the scientific level of the article.

                                                                                                               The authors

Round 2

Reviewer 1 Report

1. The authors mentioned in the response letter they have made the corrections for Suggestions 1 and 2 in the first review, but nothing was changed based on these two suggestions. It seems the authors have problems understanding the meaning of these two suggestions and the Faculty of Letters at University of Craiova did not help, either. Grammar errors can still be easily found throughout the paper; however, it will be the publisher’s job for final correcting those errors.

2. Page 1, Line 26: “there are studied the mechanical characteristics, the characteristics of water vapor transmission and the characteristics of 27 moisture absorption of Dammar films.” Should be “In [6], mechanical characteristics, characteristics of water vapor transmission and characteristics of 27 moisture absorption of Dammar films are studied.”

3. Page 3, Line 102: “The casting temperature was 21-230C.” “230C” should be “23⁰C”.

4. Page 3, Line 106-107: “The sample sizes 106 were: 250 mm long, 25 mm wide and 6.2 mm thick (according to [28]).” Please update it to “The sample sizes 106 were: 250 mm long, 25 mm wide and 6.2 mm thick based on ASTM D3039 standard [28]. The same below.

5. Page 3, Line 111: “having as reinforcement the following” should be “having reinforcement as the following”.

6. Page 4, Line 131: The title for Section 2.2 is wrong and no only the equipment is discussed in this section.

7. Please further split Section 2.2 into smaller subsections as “Tensile test”, “SEM and EDS analysis”, and “Vibration analysis”.

8. Page 5: Please remove Lines 140-143 and Lines 155-162. Mentioning the facility brand and model will be enough for a technical paper unless you are trying to sell the mentioned facility to the readers.

9. Page 5: Lines 151-152: What does “Among the types of 151 materials for which it is used there are composite materials too, with a polymer matrix.”? Looks like an excessive sentence trying to boast the multifunction of the machine. If this machine cannot be used for the planned experiment, why do you mention it here?

10. Page 5, Line 180: “an accelerometer with a 0.04 pC/ms-2 sensitivity.” Strange unit for the sensitivity of the accelerometer, could you explain what this number means?

11. Page 6, Line 193: “w1 = 0.2 m/s2 and w2 = 0.05 m/s2.” The authors mentioned in Line 187 w1 and w2 are the peak amplitude at t1 and t2. And now they have units of acceleration.

12. Page 6, Line 197: “Henceforward, by representative sample we understand the sample with the medium values of the studied mechanical properties.” Why the medium values are used to represent the material? Do you mean “medium value” or “mean value”?

13. Pages6-7, how were all the modulus of elasticity calculated? To which in these stress-strain charts did you think the deformations of materials were still linear?

14. Page 7, Line 216: I could not find the word “masic” in my dictionary.

15. Page 7, Fig. 7 caption: The label of Expoy Resin should be (d).

16. Page 8, Table 2: what can you tell from Table 2? The purpose of writing a technical paper is not simply list your experimental results, a discussion and reasoning should be included.

17. Page 8, Lines 230-231: The definition of Atomic Concentration and Weight concentration should go before Table 2 as both of them have been showed up there.

18. Page 9, Line 243: “Since in the subsection 3.2 the composite materials that are going to be studied will have the type …” should be “Since in Section 3.2 the composite materials going to be studied will have the type …”

19. Page 10, Table 5: If you are using SI throughout the paper, the unit of density should be updated to kg/m3 as well.  Also, why only cotton, flax, and hemp are included in this table but not wheat straw and cattail leaves?

20. Page 10, Lines 262-266: What is the function of defining Rp0.2 here? This parameter is only listed here for flax and cotton reinforced composite materials and not being discussed anywhere later.
21. Page 10, Lines 267-268: Please delete these two lines as they are repeated later by Line 294-297 in Page 12.

22. Pages 10-11: Why are the two curves in Figs. 10 and 11 have different slopes? Do they represent different composite materials or simply different samples?

23. Page 13, Lines 305: “… when is changed the proportion between Dammar and epoxy resin.” Should be “… when the proportion between Dammar and epoxy resin is changed.”

24. Page 350: “…we notice an increase in the damping capacity when the Dammar proportion increases in the composition.” There is no data or figures in the paper supporting the authors’ claim as damping capacity increment.

25. Page 15, Line 366: Do you mean “significant differences”?

26. Page 15, Line 367: “In the case of the loss factor, …” the parameter “loss factor” has never been defined in the paper and the authors are discussing it in the conclusion.

27. Overall, the tense used in the paper is a mess.

Author Response

Dear Reviewer 1,

In accordance with your last remarks, we have made the following changes in the article structure:

Title: “A STUDY OF SOME MECHANICAL PROPERTIES OF A CATEGORY OF COMPOSITES WITH A HYBRID MATRIX AND NATURAL REINFORCEMENTS

Authors: Marius Marinel Stănescu and Dumitru Bolcu

1. The authors mentioned in the response letter they have made the corrections for Suggestions 1 and 2 in the first review, but nothing was changed based on these two suggestions. It seems the authors have problems understanding the meaning of these two suggestions and the Faculty of Letters at University  of Craiova did not help, either. Grammar errors can still be easily found throughout the paper; however, it will be the publisher’s job for final correcting those errors.

We have contacted the editor for English grammar corrections.

2. Page 1, Line 26: “there are studied the mechanical characteristics, the characteristics of water vapor transmission and the characteristics of 27 moisture absorption of Dammar films.” Should be “In [6], mechanical characteristics, characteristics of water vapor transmission and characteristics of 27 moisture absorption of Dammar films are studied.”

We have fixed the error.

3. Page 3, Line 102: “The casting temperature was 21-230C.” “230C” should be “23C”.

We have fixed the error.

4. Page 3, Line 106-107: “The sample sizes 106 were: 250 mm long, 25 mm wide and 6.2 mm thick (according to [28]).” Please update it to “The sample sizes 106 were: 250 mm long, 25 mm wide and 6.2 mm thick based on ASTM D3039 standard [28]. The same below.

We made the change.

5. Page 3, Line 111: “having as reinforcement the following” should be “having reinforcement as the following”.

We made the change.

6. Page 4, Line 131: The title for Section 2.2 is wrong and no only the equipment is discussed in this section.

We changed the title of the subsection in “Used equipments and the analysis methods”.

7. Please further split Section 2.2 into smaller subsections as “Tensile test”, “SEM and EDS analysis”, and “Vibration analysis”.

We divided the subsection 2.2 as you require.

8. Page 5: Please remove Lines 140-143 and Lines 155-162. Mentioning the facility brand and model will be enough for a technical paper unless you are trying to sell the mentioned facility to the readers.

We have removed lines 140-143.

9. Page 5: Lines 151-152: What does “Among the types of materials for which it is used there are composite materials too, with a polymer matrix.”? Looks like an excessive sentence trying to boast the multifunction of the machine. If this machine cannot be used for the planned experiment, why do you mention it here?

We have fixed the error.

10. Page 5, Line 180: “an accelerometer with a 0.04 pC/ms-2 sensitivity.” Strange unit for the sensitivity of the accelerometer, could you explain what this number means?

The sensitivity of accelerometer is given by producer (Brüel & Kjaer). C means coulomb, pC represents picocoulomb (10-12 C), and ms-2 represents the acceleration measurement unit. The accelerometer is specifically designed for vibration measurements.

11. Page 6, Line 193: “w1 = 0.2 m/s2 and w2 = 0.05 m/s2.” The authors mentioned in Line 187 w1and w2 are the peak amplitude at t1 and t2. And now they have units of acceleration.

We made a mistake at the unit of measurement that was corrected.

12. Page 6, Line 197: “Henceforward, by representative sample we understand the sample with the medium values of the studied mechanical properties.” Why the medium values are used to represent the material? Do you mean “medium value” or “mean value”?

Since now, by a representative sample of a set, we will understand the sample for which the experimental results are close to the medium value (arithmetic mean) of the studied mechanical properties for the whole set of samples of that type.

13. Pages 6-7, how were all the modulus of elasticity calculated? To which in these stress-strain charts did you think the deformations of materials were still linear?

The modulus of elasticity is automatically given by the testing machine software. This is equal to the slope of the tangent to the characteristic curve (the tangent starting from the origin of the coordinate system).

14. Page 7, Line 216: I could not find the word “masic” in my dictionary.

Mass proportion.

15. Page 7, Fig. 7 caption: The label of Expoy Resin should be (d).

We have fixed the error.

16. Page 8, Table 2: what can you tell from Table 2? The purpose of writing a technical paper is not simply list your experimental results, a discussion and reasoning should be included.

From Table 2 we can see that the main elements that appear in the chemical structure of hybrid resin type Da 2.x, are Carbon and Oxygen.

17. Page 8, Lines 230-231: The definition of Atomic Concentration and Weight concentration should go before Table 2 as both of them have been showed up there.

We made the change.

18. Page 9, Line 243: “Since in the subsection 3.2 the composite materials that are going to be studied will have the type …” should be “Since in Section 3.2 the composite materials going to be studied will have the type …”

We made the change.

19. Page 10, Table 5: If you are using SI throughout the paper, the unit of density should be updated to kg/m3 as well.  Also, why only cotton, flax, and hemp are included in this table but not wheat straw and cattail leaves?

We have fixed the error. We have not found data on the mechanical properties of wheat straw and cattail leaves.

20. Page 10, Lines 262-266: What is the function of defining Rp0.2 here? This parameter is only listed here for flax and cotton reinforced composite materials and not being discussed anywhere later. 

For materials with a high elongation at break, we determined Rp0.2 which is the tension at which a specific remainder deformation of 0.2 % is obtained after the removal of the external load. This stress is highlighted in materials having a high elongation at which remainder plastic deformations. For materials with low elongation at break with a breakable behavior (where breakage suddenly appears) Rp0.2 not be highlighted. This tension is given by the test machine software.

To determine Rp0.2, from the axis of the specific deformations (in point of ε = 0.2% abscissa), is drawn a parallel line to the tangent at the characteristic "dilated" curve (the tangent starts by the origin of the reference system). The tension value at the point where this line intersects the characteristic curve is Rp0.2.

In Figures 10 and 11, the characteristic "dilated" curves (drawn with green color) have the scale at the top of the graph.

21. Page 10, Lines 267-268: Please delete these two lines as they are repeated later by Line 294-297 in Page 12.

We deleted these two lines.

22. Pages 10-11: Why are the two curves in Figs. 10 and 11 have different slopes? Do they represent different composite materials or simply different samples?

The "normal" characteristic curve is the one drawn in blue, and the "dilated" curve is drawn in green. The scale of the "dilated" curve is shown at the top of the graphs.

23. Page 13, Lines 305: “… when is changed the proportion between Dammar and epoxy resin.” Should be “… when the proportion between Dammar and epoxy resin is changed.”

We made the change.

24. Page 350: “…we notice an increase in the damping capacity when the Dammar proportion increases in the composition.” There is no data or figures in the paper supporting the authors’ claim as damping capacity increment.

In the paper are presented the loss factor values for the three types of hybrid resin. The damping properties increase with the loss factor increase (see explanation for the loss factor).

25. Page 15, Line 366: Do you mean “significant differences”?

We made the change.

26. Page 15, Line 367: “In the case of the loss factor, …” the parameter “loss factor” has never been defined in the paper and the authors are discussing it in the conclusion.

The damping capacity is given by the η loss factor. In the study of damped vibrations, is used the complex modulus of elasticity

E *=E(1+i η).

Under these conditions, the loss factor η is a property of material, in the same way as the modulus of elasticity, tensile strength and others. Concerning the behavior of sample on vibration, we notice an increase in the damping capacity (given by η), when the Dammar proportion increases in the composition.

27. Overall, the tense used in the paper is a mess.

We have contacted the editor for English grammar corrections.

We have marked the changes with red-colour (in the paper's content - attached to this answer).

We thank you for your viewpoints based on which we have made the necessary changes to increase the scientific level of the article.

                                                                                                               The authors

Round 3

Reviewer 1 Report

1. Page 4, Title of Section 2.2: Please update it to “Analysis methods and equipment used”. The word “equipment” is uncountable.

2. Page 6, Line 188: Typically either the “median value” or the “mean value” is used for statistical processes. Based on your explanation, please use “mean value” here to avoid any confusion.

3. Page 10, Table 5: Please mention “the authors were unable to find mechanical properties of wheat straw and cattail leaves” in the text so the readers will not think this table is incomplete.

4. Page 10, Lines 254-264: Based on the authors’ explanation, it seems the purpose of defining Rp0.2 is simply indicating flax and cotton reinforced composite materials have high elongations at break. However, this result is so obvious by checking the strain values from the stress-strain charts of composite materials reinforced by these two types of fibers. The values of Rp0.2 are simply listed in Figs. 10 and 11 and are neither discussed nor used anywhere else in the paper. Therefore, the whole Rp0.2 part can be deleted.

5.  Page 10, Line 260: What is the so called “characteristic ‘dilated’ curve”? Are the two “dilated” curves in Figs. 10 and 11 automatically generated by the tester’s software again? If so, both of them can be deleted based on Comment #4. 

Author Response

Dear Reviewer 1,

In accordance with your last remarks, we have made the following changes in the article structure:

Title: “A STUDY OF SOME MECHANICAL PROPERTIES OF A CATEGORY OF COMPOSITES WITH A HYBRID MATRIX AND NATURAL REINFORCEMENTS

Authors: Marius Marinel Stănescu and Dumitru Bolcu

1. Page 4, Title of Section 2.2: Please update it to “Analysis methods and equipment used”. The word “equipment” is uncountable.

We changed the title of subsection 2.2.

2. Page 6, Line 188: Typically either the “median value” or the “mean value” is used for statistical processes. Based on your explanation, please use “mean value” here to avoid any confusion.

We used the expression "mean value".

3. Page 10, Table 5: Please mention “the authors were unable to find mechanical properties of wheat straw and cattail leaves” in the text so the readers will not think this table is incomplete.

Before Table 5 we introduced the sentence "The authors were unable to find mechanical properties of wheat straw and cattail leaves".

4. Page 10, Lines 254-264: Based on the authors’ explanation, it seems the purpose of definingRp0.2 is simply indicating flax and cotton reinforced composite materials have high elongations at break. However, this result is so obvious by checking the strain values from the stress-strain charts of composite materials reinforced by these two types of fibers. The values of Rp0.2 are simply listed in Figs. 10 and 11 and are neither discussed nor used anywhere else in the paper. Therefore, the whole Rp0.2 part can be deleted.

We deleted the explanation for Rp0.2.

5.  Page 10, Line 260: What is the so called “characteristic ‘dilated’ curve”? Are the two “dilated” curves in Figs. 10 and 11 automatically generated by the tester’s software again? If so, both of them can be deleted based on Comment #4.

In Figures 10 and 11 we removed the "dilated" curves.

We have marked the changes with red-colour in the paper's content - attached to this answer.

We thank you for your viewpoints based on which we have made the necessary changes to increase the scientific level of the article.

The authors

This manuscript is a resubmission of an earlier submission. The following is a list of the peer review reports and author responses from that submission.

Round 1

Reviewer 1 Report

This paper investigated two mechanical properties of the bio-composites that made from various natural fiber reinforcements using Dammar vegetable resin. The authors described the starting materials used for making experimental samples, the equipment utilized in their experiments, and then followed by a discussion of their final results. However, this paper is not well presented with a strange organization, with many important information and details missing. It needs a major revision with additional details and explanations before it could be reconsidered for publication. Comments are listed as follows:

Major Issues:

1.      What are the main purposes of writing this paper? The abstract only mentioned what the authors studied from other researchers’ work and what they did for this paper, while the introduction part is completely a literature review section. The authors need to mention why it is important to carry out the work they have done and where could the research results of this paper be applied to.

2.      Page 2, Section 2.1, Paragraph 1: Why were the volume proportions of 55%, 65%, and 75% used for the Dammar resin in the bio-composites? Are there any standards to follow or were these numbers picked randomly?

3.      In the same paragraph: What reinforcement was used in the first stage of the experiment? Without mentioning the exact reinforcement fiber used in the mechanical tests, all the obtained data will be meaningless.

4.      For bio-composite samples made for Stages 1 and 2: Why was the dimension of 200 mm long, 20 mm wide adopted for making test samples? Is this dimension from the standard the authors used?

5.      Page 3, Section 2.2: The full title for “SR EN ISO 6892-1:2016” standard is “Metallic materials Tensile Testing”, which should not be used for bio-composite material tests. Also, the standard mentioned in the paper “SR EN ISO 6892-1:2010” could not be found online. The most updated 2016 version simply replaces the 2009 but not 2010 version.

6.      Page 4, Paragraphs for the vibration analysis: Please state why the vibration properties of the bio-composite material are particularly important in this case as most of the studies dealing with composite materials only study the tensile, flexural, and shearing properties.

7.      Page 4, Paragraph 3: The experimental method described is far from clear enough. Please explain the experimental setup in detail and why such a setup was used (any standard to follow?). A picture similar to Figure 2 will be very helpful. Also, why were the 4 different free lengths from 120 mm to 180 mm used for the vibration test?

8.      Figure 2 should be placed in Section 2.2 instead of Section 3 where the test results are presented.

9.      Page 6, what is the purpose of the EDS analysis? Why is the elemental composition of the bio-resin sample important in this study? Since EDS does not deal with mechanical properties of the material, it will be a conflict to have this part if the current title of the paper is not changed.

10.  Page 11, Section 3: Please put all the discussion parts (especially the calculation of the loss factor) into Section 3 and make it the “Results and Discussion” part. Section 4 should be your conclusion section with a summary of what has been found through this study and their contributions to the current engineering society and possible applications.

11.  The references cited in this paper are really old, with a few of them over 20 years old. Please re-do the literature review and update. Here are some articles that could be cited by this paper:

(1) Vaisanen, T., Das, O., Tomppo, L., 2017. A review on new bio-based constituents for natural fiber-polymer composites. J. Clean. Prod. 149, 582-596.

(2) L. Jiang, D. Walczyk, G. McIntyre, R. Bucinell, G. Tudryn, Manufacturing of biocomposite sandwich structures using mycelium-bound cores and preforms,” J. Manuf. Pro., vol. 28, no. 1, pp. 50-59, Aug. 2017.

(3) L. Jiang, D. Walczyk, and G. McIntyre, “A new approach to manufacturing biocomposite sandwich structures: investigation of preform shell behavior,” J. Manuf. Sci. Eng., vol. 139, no. 2, pp. 021014.1-11, Feb. 2017, doi:10.1115/1.4034278.

(4) Sachs, A., and Netravali, A., 2012, “Starch Based Resins and Their Composites From Paper or Natural Fibers,” Last accessed Oct. 23, 2015.

(5) Lee, K. Y., Shamsuddin, S. R., Fortea-Verdejo, M., and Bismarck, A., 2014, “Manufacturing of Robust Natural Fiber Preforms Utilizing Bacterial Cellulose as Binder,” J. Vis. Exp., 87, p. e51432.

Minor Issues:

1.      Please avoid using the first person in a research article as to make it professional

2.      Abstract: Please mentioned the full name of EDS the first time it shows up in the article

3.      The citation numbers should be in order in the context of the paper

4.      Page 1, Paragraph 2: the first sentence is incomplete, while the second half of the paragraph is really hard to understand. Please break it into shorter sentences.

5.      Page 2, Line 76: “having … as reinforcement:” Missing something here.

6.      Page 3, Lines 80 and 82: the units for both specific masses are incorrect.

7.      Page 3, Line 88: “…, 20 wide.” Missing unit.

8.      Page 3, Figure 1: please explain all the marks on the samples so the readers know which sample is made from what material.

9.      Page 3, Lines 96 and 98: Please update the decimal points from “,” to “.”. There are same mistakes at other places of the paper.

10.  Please make the unit for elastic modulus consistent throughout the paper and ‘MPa” is recommended instead of “N/mm2”.

11.  Page 4, Lines 112-120: No need to list all the properties of the SEM.

12.  Page 5, Line 153: What is “Da 2.x si”?

13.  Page 7, Lines 171-173: Why the Dammar volume proportion of composite material in Section 3.2 changed to 55%? Isn’t the volume proportion of sample Da 2.x set 65%? Please specify.

14.  Page 7, the equation for calculating the damp factor: need to cite the reference for it.

15.  Page 7, Line 178: “… Dammar samples having the length of 20 mm.” Is this a new set of samples with a different length?

16.  Page 10, Figure 13: please be consistent about the information in the figures. The elongation value A is only showing up in Figure 13 but not in Figures 9-12.

17.  Page 11, Line 249: “… had a low elongation at break, of almost 10%.” I don’t see any elongation value from your experiment results that are approaching 10%.

18.  Page 11, Line 261: “… which begins to yield, the stress transferring to the fibers.” Sentence incomplete.

19.  Page 12, Line 269: What is the “intermediate character”? Please specify in detail and also point it out in the stress-strain chart if possible.

20.  Page 12, Line 273: What is the variable “v” in the loss factor equation? Also, this equation needs to be cited.

21.  Page 12, Line 283: “If we compare the damping factor value of the 65% Dammar resin ...” So the Dammar resin got a damping factor by itself?

22.  Page 12, Line 289: “… we obtain materials that have a tensile strength and a modulus of elasticity three times higher than those of the bio-resin.” I could not find any information regarding the tensile strength and elastic modulus of the bio-resin in the paper, please specify those values for the resin only and then compare.

23.  Pages 12-13: Both Tables A1 and A2 missing units.

Reviewer 2 Report

Basic work, just comparing various materials, without scientific approach.

No discussion of results.

More comments see in attachment.
